# Topical Ocular Administration of Progesterone Decreases Photoreceptor Cell Death in Retinal Degeneration Slow (rds) Mice

**DOI:** 10.3390/ph15030328

**Published:** 2022-03-09

**Authors:** Adrián M. Alambiaga-Caravaca, Antolín Cantó, Vicent Rodilla, María Miranda, Alicia López-Castellano

**Affiliations:** 1Departamento de Farmacia, Facultad de Ciencias de la Salud, Instituto de Ciencias Biomédicas, Universidad Cardenal Herrera, CEU Universities, C/Santiago Ramón y Cajal, s/n., Alfara del Patriarca, 46115 Valencia, Spain; alacaradr@gmail.com (A.M.A.-C.); antolin.cantocatala@uchceu.es (A.C.); vrodilla@uchceu.es (V.R.); 2Departamento de Ciencias Biomédicas, Facultad de Ciencias de la Salud, Instituto de Ciencias Biomédicas, Universidad Cardenal Herrera, CEU Universities, C/Santiago Ramón y Cajal, s/n., Alfara del Patriarca, 46115 Valencia, Spain

**Keywords:** progesterone, retinitis pigmentosa, topical administration, in vivo, ocular formulations

## Abstract

Retinitis pigmentosa (RP) is an inherited eye disorder which triggers a cascade of retinal disorders leading to photoreceptor cell death and for which there is currently no effective treatment. The purpose of this research was to study whether ocular administration of a solution of progesterone (PG) in β-cyclodextrins (CD) could delay photoreceptor cell death and counteract the gliosis process in an animal model of RP (rds mice). The possible effect of PG reaching the contralateral eye through the circulatory system was also evaluated. Finally, this research discusses and evaluates the diffusion of the drug from possible topical formulations for ocular administration of PG. A group of rds mice received one drop of a solution of PG in CD every 12 h for 10 days to the left eye, while the right eye was left untreated. Another group of rds mice (control) received the drug vehicle (PBS) on the left eye and, again, the right eye was left untreated. Once the treatment was finished on postnatal day 21, the animals were euthanized and histological immunofluorescence studies (TUNEL, GFAP, and DAPI staining) were carried out. Our results showed that the administration of a solution of PG in CD (CD-PG) as drops significantly decreased cell death and inflammation in the retina of the PG-treated eyes of rds mice. No effect was seen in the contralateral eye from PG that may have entered systemic circulation. In conclusion, CD-PG applied topically as drops to the eye decreases photoreceptor cell death in the early stages of RP, delaying vision loss and decreasing gliosis.

## 1. Introduction

Retinitis pigmentosa (RP) is a group of inherited diseases that produce progressive degeneration of photoreceptor cells advancing eventually towards complete blindness. Rods are the first photoreceptor cell type affected by RP, and this results in night blindness and the reduction of the visual field. The death of rod cells leads to a pathological environment in the retina that affects cone photoreceptors and induces the death of the latter cells. In the final stages of this disease, electroretinograms show that the activity of the retina decreases, there is no excitation of the photoreceptors, and the accumulation of pigment deposits (clusters) can be seen in the peripheral retina [1].

One in 4000 people is affected by RP, which means there are almost two million people throughout the world affected by this disease. Although RP is a rare disease, it is the most common cause of inherited blindness. No effective treatment for RP exists, and because a heterogenic group of over 100 genes are involved in RP, it is hugely difficult to find an efficient gene therapy treatment for this disease [1,2]. Among the autosomal dominant forms of RP, which constitute 20% to 25% of the non-syndromic form of RP, two forms of mutations are the most common in humans: mutations in the rhodopsin and the peripherin (PRPH2/RDS) genes [3]. Mutations in the human PRPH2/RDS gene have been identified as the cause of disease in over 150 individuals [4]. The retinal degeneration slow (rds) mice, which have a mutation in the PRPH2 gene, have been used as a model in the present study because the rate of cell death in these mice closely resembles that observed in RP-affected humans and in other forms of progressive neuronal degeneration [5].

Different therapeutic approaches have been adopted to halt photoreceptor degeneration by pharmacological treatments. It has been suggested that long-term protection against oxidative damage may delay cone death in RP patients, regardless of the causative mutation [6]. It has been reported that PG alone or in combination with other substances, administered orally, can slow down photoreceptor cell death in *rd10* and *rd1* mice (retinal degeneration (rds) experimental models for RP) by reducing oxidative stress and decreasing the levels of inflammatory cytokines while increasing the amounts of antioxidant molecules [7,8,9,10,11,12].

Orally administered PG can be inactivated rapidly, as it undergoes extensive hepatic and intestinal metabolism; its plasma half-life is 25 min. PG (log P_o/w_ = 3.9) is considered practically insoluble in isotonic buffer solution (7–10 µg/mL) [13,14], it dissolves slowly and incompletely in gastrointestinal fluids, and has rapid hepatic metabolism. These characteristics make the bioavailability of orally administered PG rather limited, thus hindering its therapeutic administration. High concentrations of orally-administered PG are needed for a small amount to reach the eye [14]. Therefore, topical administration of PG directly onto the eye surface may be an interesting alternative to reduce the concentration which needs to be administered. However, although it is known that oral PG administration at high doses can reach photoreceptors, it is not known whether it would affect neuroretinal cell death [7,15].

Our group has investigated the incorporation of PG into different topical ocular formulations: solution, micelles, and inserts. These formulations were intended for corneal and/or scleral administration. The diffusion of PG from these formulations through the cornea and sclera into the eye has been studied in ex vivo experiments using rabbit eyes. The drops formulated and used for the trans-corneal and trans-scleral diffusion studies consisted of a solution of PG incorporated in β-cyclodextrins (CD-PG) [16]. It is interesting to note the use of cyclodextrins in drops can increase their solubility and increase drug permeability [17,18,19]. Other PG ocular formulations developed by our group were polymeric micelles of Soluplus (20%) and Pluronic F68 (20%), both of them loaded with pure PG [20]. Polymeric micelles were used for their ability to encapsulate hydrophobic drugs, facilitating the development of formulations with poorly soluble active ingredients, such as PG. Our previous research results showed that encapsulation in those polymeric micelles facilitated the development of aqueous formulations by increasing the permanence on the ocular surface and increasing the penetration of the molecule into deeper ocular structures as indicated by other authors [18,20,21,22]. The other formulation developed by our group was a polymeric eye insert composed of 59% polyvinyl alcohol, 39% polyvinylpyrrolidone K30, and 2% propylene glycol that was loaded with CD-PG [16]. The advantages of eye inserts compared to liquid formulations are numerous. The drug-loaded insert is designed to adhere to the conjunctiva or directly to the cornea, allowing longer residence time and bioavailability [23]; it allows a more precise dosage with controlled release, it has minimal systemic absorption, and would reduce administration frequency [24]. Several studies have shown that PG could be a promising agent for the treatment of retinal disorders after oral administration [7,9,25], whereas in the present work we shall explore the possibility and effectiveness of local ocular administration of PG.

The following are the objectives of the experimental research carried out and presented in this study: (1) to assess the effect of the CD-PG solution in delaying photoreceptor cell death; (2) to study the effect of the CD-PG solution in the gliosis process in the rds mouse model; (3) to study the possible effect of PG reaching the contralateral eye through the circulatory system; and additionally, (4) to discuss and evaluate the diffusion of the drug from possible topical formulations and their suitability for the ocular administration of PG.

## 2. Materials and Methods

### 2.1. Experimental Design

rds mice were housed in the facilities of the Research Unit at CEU Cardenal Herrera University. The animals were kept in cages at 25 °C, 60% humidity, and standard conditions of dark and light cycles (12 h) with free access to food and water (Harlan Ibérica S.L., Barcelona, Spain). The handling and care of the animals were approved by the CEU Cardenal Herrera Universities Committee for Animal Experiments (reference 2021/VSC/PEA/0108) and were also performed following the ARVO (Association for Research in Vision and Ophthalmology) statement for the use of animals in Ophthalmic and Vision Research.

In the present work, in vivo studies were performed on rds mice by administering CD-PG solution as eye drops. The PG eye drops (1 mg/mL) were prepared using PG in methyl-β-cyclodextrin complex powder (Sigma Aldrich Chemical, Co. St. Louis, MO, USA) (85.2 mg PG/g) which was reconstituted in water. The drops were administered twice daily (every 12/h). The day the mice were born was marked as postnatal day 0 (PN0). The mice were separated in two groups to receive CD-PG treatment (PG-treated mice; *n* = 10) or to serve as controls (control mice; *n* = 8) (Figure 1). From PN11 to PN20 (both inclusive) (Figure 1C), the left eyes of the PG-treated mice or control mice received either a drop of CD-PG solution or a drop of 0.1 M phosphate buffer saline (PBS) pH 7.2 every 12 h (Figure 1). The right eyes of the mice were not treated with anything, leaving them as control eyes. PG-blank (Figure 1B) would serve to evaluate the possible effects derived from CD-PG administration to the contralateral eye (CD-PG-T) (Figure 1B).

### 2.2. Histological and Immunofluorescence Studies

At PN21 the mice were euthanized, the eyes were enucleated, and after puncturing the corneas with a needle, the eyes were submerged in 4% paraformaldehyde for 2 h. The eyes were then washed three times in PBS solution for 10 min and submerged overnight at 4 °C in a PBS solution containing a final sucrose concentration of 0.3 M. The eyes were transferred to a PBS solution with 0.9 M sucrose 24 h later. The eyes were embedded in Tissue Tek (Sakura Europe, Spain), and 8 µm sections were obtained using a Leica CM 1850 UV Ag protect cryostat, (Leica Microsistemas SLU, Barcelona, Spain). The sections were transferred to superfrost slides (Thermo Fisher Scientific, Braunschweig, Germany) and kept at −20 °C [7,8].

A double immunostaining process was carried out to detect the dying cells using a terminal deoxynucleotidyl transferase nick end labelling assay (TUNEL) and to detect glial fibrillar acid protein expression (GFAP). The TUNEL assay identifies and quantifies cellular death, whereas upregulation of GFAP expression is used as a marker of reactive gliosis, which appears as a response to retinal injury. Tissue sections were washed three times with PBS 1% and triton 0.1% solution for 10 min. Unspecific epitopes were blocked by incubating the tissue sections in 1% PBS and 5% normal goat serum solution for 1 h at 4 °C. The sections were then washed with 1% PBS and the TUNEL in situ cell death kit (Roche Diagnostics, Mannheim, Germany) was used, as previously reported [7]. The sections were incubated at 4 °C overnight in a solution of GFAP antibody in 1% PBS (1:500, Dako cytomation, Denmark). The primary antibody was detected with the fluorescence-conjugated secondary antibody Alexa Fluor 568 (Invitrogen, Life Technologies, Madrid, Spain). Finally, the sections were mounted using Vectashield^®^ mounting medium with DAPI (Vector, Burlingame, CA, USA) to provide nuclear counterstaining.

Images of the retina were taken at 20× magnification with a Nikon DS-Fi1 camera attached to a Leica DM 2000 microscope. The software used for the image capture was the application Suite version 2.7.0 R1 (Leica Microsystems SLU, Barcelona, Spain). To ensure that the area of the retina pictured was the same in all sections, photos were taken using the optic nerve as a positioning reference. The dying cells were counted, divided by the area of the retina selected, and multiplied by 10^6^ to obtain integer numbers. GFAP was quantified as the percentage of the labelled space compared to the whole retina. Finally, the number of rows in the outer nuclear layer (ONL), which would correspond to the nucleus of photoreceptors, were counted using the DAPI counterstaining of cell nuclei. These quantifications were carried out with the help of the software ImageJ Fiji.

### 2.3. PG Permeability Coefficients

To compare the topical formulations for ocular administration of PG and CD-PG we analyzed the trans-corneal and trans-scleral permeability of PG and CD-PG from different ocular formulations studied by our research group (CD-PG solution, PG micelles, and CD-PG insert) [16,20]. The apparent permeability coefficients (P_eff_) were recalculated using Equation (1) [26] using the data obtained from ex vivo experiments with rabbit corneas and scleras from solution, micelles, and polymeric PG-embedded insert. This equation considers a continuous change in the donor and acceptor concentrations and is valid under both sinking and non-sinking conditions.
(1)$$C_{\mathrm{receiver},\;t\;}=\frac{Q_{\mathrm{total}}}{V_{\mathrm{receiver}}{+V}_{\mathrm{donor}}}+\left[\left(C_{\mathrm{receiver},t-1}\cdot f\right)-\left(\frac{Q_{\mathrm{total}}}{V_{\mathrm{receiver}}{+V}_{\mathrm{donor}}}\right)\right]{\cdot e}^{P_{\mathrm{eff}}\cdot S\cdot \left(\frac{1}{V_{\mathrm{receiver}}}+\frac{1}{V_{\mathrm{donor}}}\right)\cdot \Delta t}$$

C_receiver_, t is the PG concentration (μg/mL) in the receptor compartment at time t, Q_total_ is the total amount of PG in the insert, V_receiver_ is the volume in the receptor compartment, V_donor_ is the volume in the donor compartment, C_receiver,t−1_ is the amount of PG in the receptor compartment at a previous time, f is the replacement dilution factor of the sample, S is the surface area of the membrane, and Δt is the time interval. The curve fittings were performed by a non-linear regression, minimizing the sum of the squared residuals.

### 2.4. Statistical Analysis

The Shapiro–Wilk test was used to assess the normality of the histological and immunofluorescence data and the Levene test was used to check for homoscedasticity. The analysis of variance (ANOVA) test and the Bonferroni test were used to compare the groups. A statistical analysis for permeability coefficients was performed using the Mann–Whitney test between the values obtained for both membranes (cornea and sclera) and each formulation. The analyses were carried out using the software IBM SPSS v27 with α = 0.05. The results are presented as means ± SD.

## 3. Results

The topical CD-PG solution treatment of the eye showed a statistically significant difference in the amount of photoreceptor cell death. Figure 2 shows how CD-PG treatment diminished the amount of TUNEL-labelled cells per area in the PG-T eye compared with the control, which received PBS. Furthermore, the study showed a statistically (*p* < 0.05) significant decrease in TUNEL-positive cells in PG-T in contrast to PG-blank in the same mice but not between the control and blank eyes. In addition, there were no statistical differences between the number of TUNEL cells in the retina of PG-blank, control, and blank eyes. These results showed a decrease in the photoreceptor cell death in the CD-PG-T eye and also prove that CD-PG did not have any effect on the CD-PG-blank eye. These data also demonstrated that topically applied CD-PG protected photoreceptors and delayed cell death in this animal model. In addition, we demonstrated that CD-PG applied in one eye did not affect the non-treated contralateral eye, suggesting that there was no effect derived from topically administered CD-PG on the contralateral eye.

The results related to GFAP expression in the retina were similar to those obtained by TUNEL staining, showing that there was an important decrease in the PG-T GFAP staining and a statistically significant difference in GFAP expression between PG-T and the other eyes (Figure 3). When GFAP expression between control, blank and PG-blank eyes was compared, no statistical differences were found. In addition, there was a decrease in the amount of GFAP expression in CD-PG-T but not in PG-blank, as was reported for TUNEL staining. As before, these results also show that there was no effect from CD-PG treatment in the contralateral eye (PG-blank) to that receiving CD-PG (CD-PG-T).

Finally, the results obtained by counting the number of cell rows in the ONL, as a means to evaluate photoreceptor cell death and retinal degeneration, did not show any statistical differences between CD-PG-T, CD-PG-blank, control, and blank (Figure 4). A possible explanation for not observing a significant decrease in the amount of ONL cell rows could be that although the maximal amount of cell death in the rds mouse model has been reported to occur between PD17 and PD21 [27], in our case, cell death may have occurred beyond 21 days after birth, which has also been reported by other research groups [28,29].

Recently, our group has studied different formulations of PG for topical delivery to the eye, including eye solution, inserts with CD-PG and micelles of Soluplus^®^ and Pluronic^®^ with pure PG [20]. To select the best ocular formulation for the topical administration of PG, all the results obtained in ex vivo diffusion PG and CD-PG studies carried out in rabbit eyes have been re-analyzed. In the diffusion experiments previously performed with the inserts, the concentration of CD-PG in the donor compartment decreased with time [16]. The PG in the inserts was less than 90%, exactly 55% and 30% when the membranes were cornea and sclera, respectively. We considered it was less accurate to calculate permeability coefficients assuming constant concentrations in the donor compartment. Therefore, to compare all the formulations tested, Equation (1) (see Section 2) was used to re-calculate the P_eff_ with all the data obtained in the previous studies performed with drops and micelles [13,14]. This equation considers that there is a continuous change in the concentrations in the donor compartment, is valid under sink or non-sink conditions, and allowed us to compare all the data obtained in the previous studies. Table 1 shows the permeability values from the original references (sink conditions) and the recalculated data (sink and non-sink conditions). The P_eff_ obtained with this mathematical treatment did not differ statistically from the value obtained by assuming that the concentration in the donor compartment remained constant (*t*-test; *p* > 0.05).

Figure 5 shows the P_eff_ of PG calculated with the different PG and CD-PG formulations through the corneas and scleras of rabbits. These formulations were prepared with CD-PG (solution and inserts) and micelles of Soluplus and Pluronic with pure PG.

Ocular formulations of PG showed significant differences in P_eff_ values (Figure 5). On the one hand, comparing liquid formulations, solution and micelles, solution showed higher permeability compared to micelles in the sclera (*p* < 0.05). Polymeric micelles are capable of encapsulating hydrophobic drugs such as PG, and the amphiphilic excipients used in micelles can facilitate the permanence of the formulation on the ocular surface compared to solution and can also inhibit efflux pumps in the corneal tissue [30]. Micelles can facilitate drug penetration through the cornea and sclera, providing therapeutic drug levels at the back of the eye [22,31]. Pluronic micelles provided higher permeability coefficients than Soluplus micelles, which may be related to their size of a few nanometers [20]. However, the polymeric micelles studied did not provide higher PG permeability than solution in the sclera, and only Pluronic micelles provided higher permeability coefficients than solution in the cornea (*p* < 0.05). This may be explained by the presence of cyclodextrins, which have been shown to increase drug permeability [19]. Both solutions and micelles have the disadvantage that they remain in contact with the ocular membrane for a short time before they are completely removed by tear production and blinking. Approximately 90% of drug formulations for ophthalmic administration are eye drops, mainly due to their ease of administration and excellent acceptance by patients [32]. In contrast, the bioavailability of drugs administered topically in drops is less than 5% [33,34,35], as defense mechanisms, such as lacrimation or tear turnover, are activated, which dilute the drug and remove it from the ocular surface. Excess volume is drained through the nasolacrimal duct into the systemic circulation [1,6,36]. Furthermore, ocular solutions are primarily intended for the treatment of afflictions of the anterior segment of the eye.

## 4. Discussion

The main objective of RP treatments, including CD-PG treatment, is to preserve photoreceptors and avoid, or at least decrease, the photoreceptor cell death rate. In this sense, TUNEL staining, which labels cells that have started apoptotic cell death, confirm the capability of CD-PG topical treatment to partially reverse photoreceptor cell death (Figure 2B) (*p*-value < 0.05). These data reveal that a higher amount of photoreceptor cells remain in the neuroretina and that the photoreceptor cell death rate decreases with CD-PG topical treatment. Orally administered PG to *rd1* and *rd10* RP mouse models has shown that PG can delay photoreceptor cell death. In the case of *rd10* mice, orally administered PG decreased the number of TUNEL-positive cells by 50% in three studied areas of the retina (far periphery, mid periphery, and central retina) but only in the far periphery was there a statistically significant difference [7]. Our data show a similar reduction in TUNEL expression (45.21%), thus demonstrating that topically applied CD-PG can be as effective in decreasing cellular death as orally administered PG. The possible clinical implications of these findings are obvious if PG can be administered as effectively by ocular topical administration, eliminating both first pass metabolism, which is known to be extensive, and the need for systemic distribution of the drug.

In the retina, the most well-known gliosis marker is the upregulation of GFAP expression by Müller cells [37,38]. Gliosis develops as a response to any kind of damage to the nervous system, including the retina. The recruitment of glial cells to the damaged area has been demonstrated to occur in several retinal degenerative diseases, such as RP, diabetic retinopathy, glaucoma, and retinal detachment, and in several mouse models. This reactive gliosis, which occurs upon retinal injury, can have detrimental effects on vision, as the proliferation of these glial cells on the retina frequently results in substantial vision deterioration, which may even lead to blindness in serious cases. Previous results using orally administered PG in the *rd10* mouse model have reported no differences in GFAP expression in PG-treated and untreated mice [7], although other studies using the *rd1* mouse model, such as those carried out by Ramirez-Lamelas [8] and Sanchez-Vallejo [15], have reported that PG treatment can delay reactive gliosis. Similarly, our data show that topical treatment with CD-PG partially inhibits GFAP expression (Figure 3).

Experimental data evaluating the number of ONL nuclear rows after oral PG administration in *rd1* mice have shown that the treatment does not have any effect at the first studied age (PN11), but preservation of ONL nuclear rows is seen later in time (PN13 and PN15) [15]. Our results did not show any effects on ONL, although it remains to be seen whether the effects of ocularly applied CD-PG could have had an effect if ONL had been evaluated in a longer time frame.

As there were no statistical differences between the data obtained in the contralateral eye and the one receiving CD-PG treatment (blank CD-PG) and those receiving the vehicle (control) or no treatment (blank), it can be concluded that CD-PG local treatment in one eye does not have an effect in the contralateral eye, suggesting that it does not cross the blood–retinal barrier or distribute sufficient CD-PG through the systemic circulation to trigger a detectable response.

CD-PG inserts showed lower permeability compared to solution and micelles (Figure 5), but the advantages of eye inserts compared to liquid formulations are numerous. The main advantage of inserts is that they increase the contact time of the formulation, increasing drug availability and allowing controlled release. Thus, effective drug concentration to the eye can be ensured for a longer period than with drops. Drug dosing is also more precise, and the risk of systemic side effects is reduced. In addition, the inserts have a longer shelf life and the presence of additives, such as preservatives, is not necessary [39]. The constant contact of the insert with the ocular membranes suggests that it would be capable of transporting PG in greater amounts to the neuroretina, albeit more slowly than solution or micelles. This has been demonstrated in ex vivo biodistribution studies in pig eyes performed by our research group, with different formulations of CD-PG (drops and corneal and scleral inserts of different sizes) at the same initial concentration. They were administered topically to the ocular surface of whole pigs’ eyes for 8 h. The PG levels were quantified by UHPLC in various ocular tissues. In the neuroretina, thePG concentration was 2.63 ± 1.08 µg/g of tissue for the solution of CD-PG and 23.19 ± 10.55 µg/g of tissue from one of the inserts [data pending publication]. Furthermore, in that study, the relationship between increased CD-PG concentration in the formulation and increased amounts of PG achieved in the neuroretina was demonstrated, so there is a correlation between the concentration in the insert and the amount of PG reaching the neuroretina. Therefore, although in the in vivo studies described in the current paper we have resorted to CD-PG solution because of the small size of eyes in the newborn mice used as a model, CD-PG inserts will remain nevertheless, a very interesting ocular formulation for topical application of PG.

## 5. Conclusions

The results shown here demonstrate that CD solution applied topically in drops to the eye has a beneficial effect on delaying photoreceptor cell death and gliosis in the rds mouse model. Furthermore, the lack of any effect on the CD-PG-blank shows that there is no detectable effect from the topically administered CD-PG to one eye on the contralateral one. These results show the feasibility of using topically administered CD-PG to the eye as a possible treatment for retinal degeneration. Although the CD-PG solution provided greater permeability through the sclera and the Pluronic F68^®^ micelles through the cornea, the insert is still considered to be the best ocular formulation tested, as it releases PG in a controlled manner. From a clinical point of view, the possibility of using topical ocular CD-PG administration has important implications and advantages, but further studies in larger animals are warranted. The use of cyclodextrins increases the solubility of an otherwise rather insoluble molecule, thus allowing the preparation of aqueous formulations, instead of having to use alcohols or other solvents, which could be detrimental to the cornea and other eye structures. Ocular CD-PG administration enables the drug to reach the target tissue directly, in this case the neuroretina, without having to resort to systemic distribution of the drug. As opposed to oral administration, ocular administration of PG eliminates first-pass metabolism, which in the case of progesterone is rather extensive, thereby allowing the use of smaller doses of the drug and increasing its effectiveness.

## Figures and Tables

**Figure 1 pharmaceuticals-15-00328-f001:**
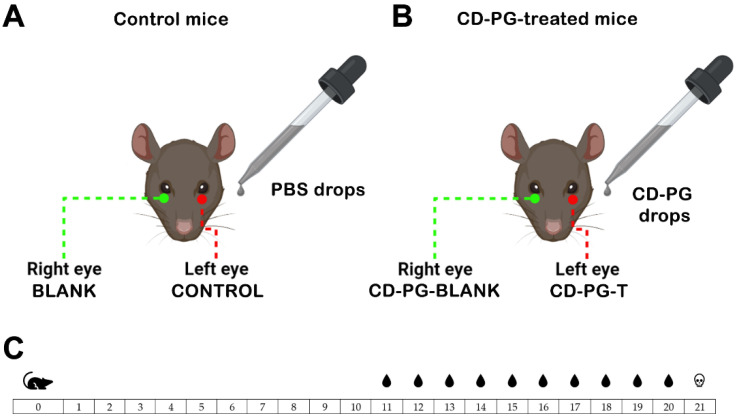
Scheme of the experimental design. (**A**) Control mice: the left eye received a drop of PBS (control) every 12 h, whereas the right eye was left untreated (blank). (**B**) CD-PG-treated mice received a drop of a 1 mg/mL CD-PG to the left eye (PG-T), whereas the right eye (PG-blank) was left untreated to evaluate the possible effects caused by CD-PG administered to the contralateral eye. (**C**) Scheme of the treatment protocol from the time mice were born (day 0) to the time mice were sacrificed (day 21), showing the days the mice received treatment (days 11–20).

**Figure 2 pharmaceuticals-15-00328-f002:**
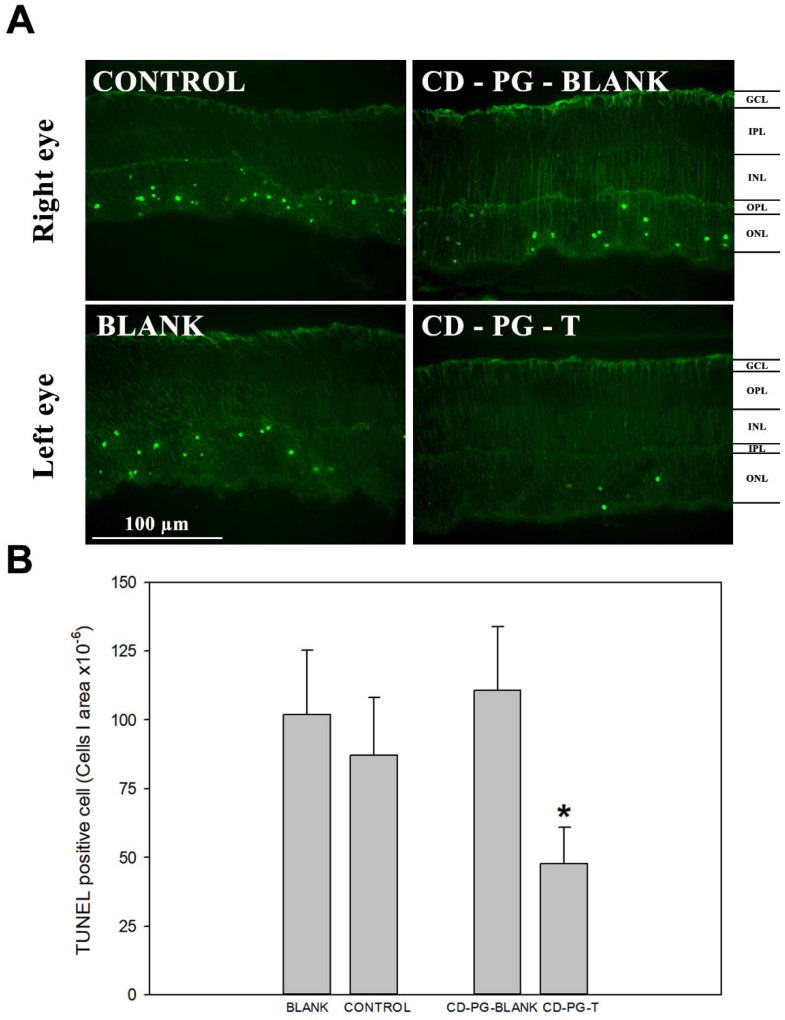
TUNEL immunofluorescence. (**A**) An example of TUNEL staining for each experimental group. (**B**) Results of TUNEL quantification. *: Statistically significant difference (*p*-value < 0.05) of CD-PG-TE vs. control, blank, and CD-PG-blank.

**Figure 3 pharmaceuticals-15-00328-f003:**
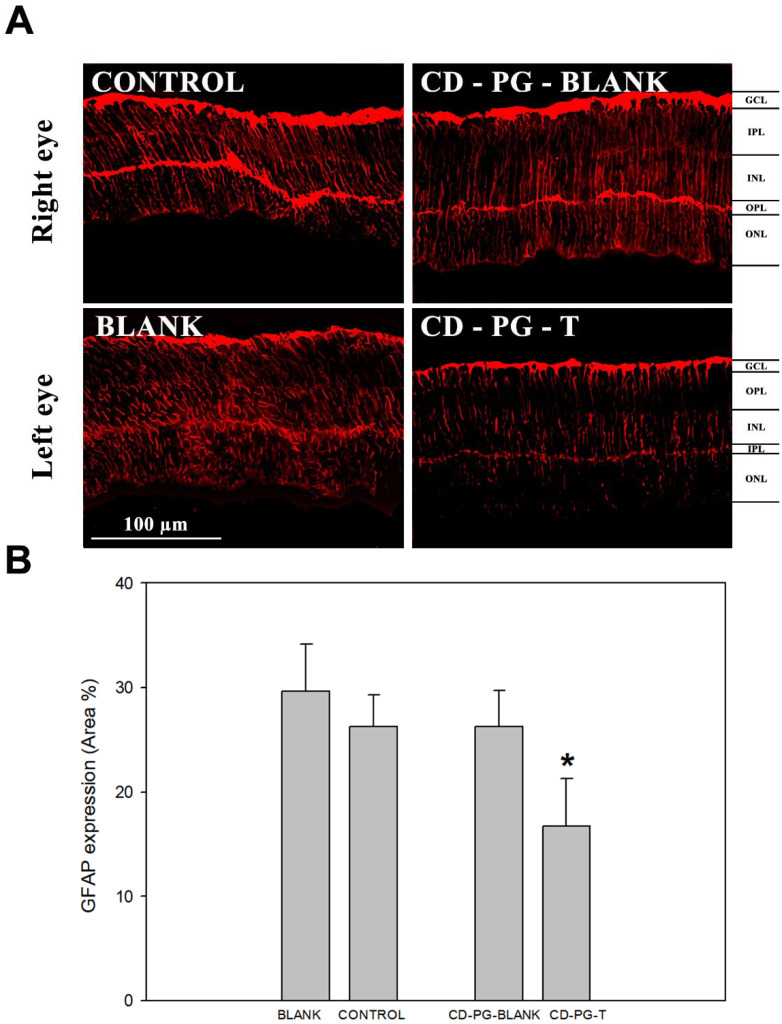
GFAP immunofluorescence. (**A**) An example of GFAP staining for each experimental group. (**B**) Results of GFAP quantification. * Statistically significant difference (*p*-value < 0.05) of CD-PG-T vs. control, blank, and CD-PG-blank.

**Figure 4 pharmaceuticals-15-00328-f004:**
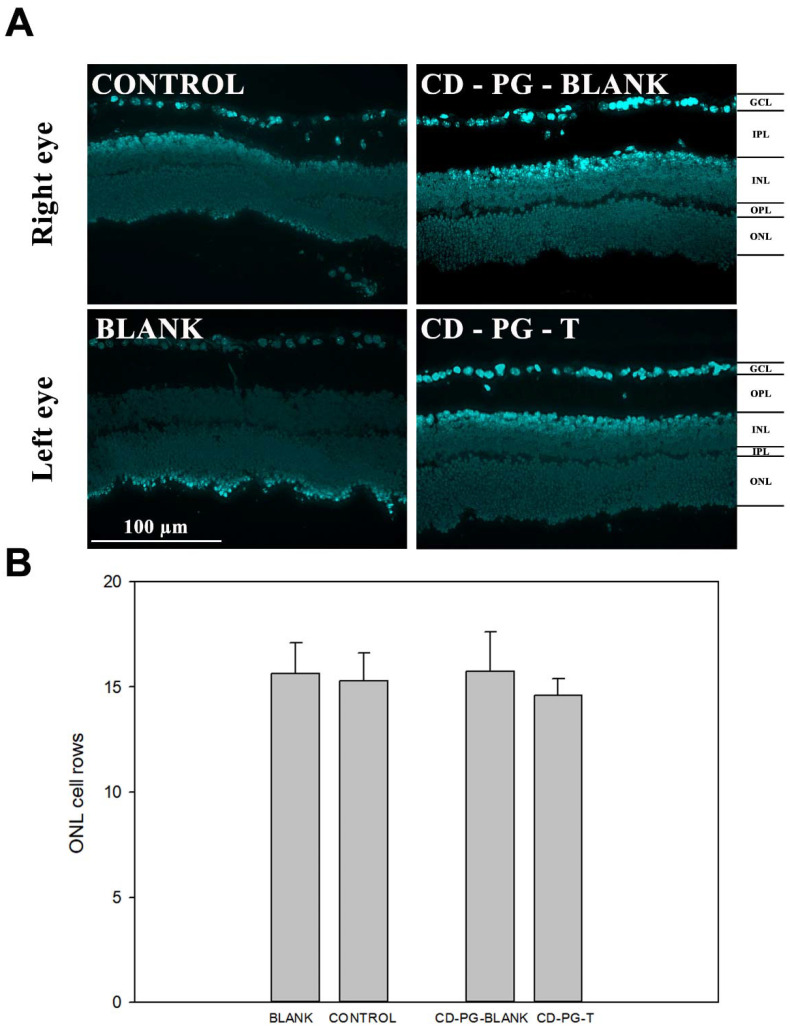
DAPI staining. (**A**) An example of DAPI staining for each experimental group. (**B**) Results of DAPI quantification in the outer nuclear layer (ONL).

**Figure 5 pharmaceuticals-15-00328-f005:**
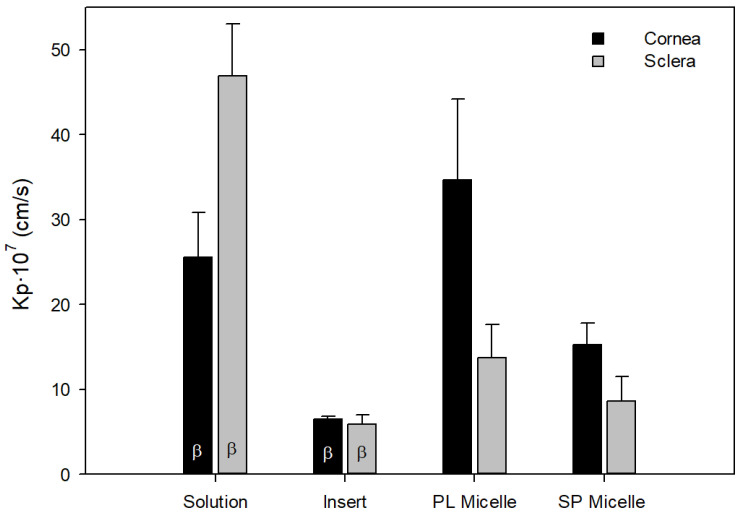
Apparent permeation coefficients (P_eff_, cm/s) of PG in the corneas and scleras of rabbits were obtained with different PG topical formulations. β indicates that those formulations were prepared as CD-PG (solution and inserts), whereas micelles of Soluplus and Pluronic were prepared with pure PG.

**Table 1 pharmaceuticals-15-00328-t001:** Permeability values from original references (^a^ [16] and ^b^ [20]) and those recalculated using sink and non-sink conditions.

Formula Conditions	CD-PG Solution		PG PL-Micelle		PG SP-Micelle	
	Cornea	Sclera	Cornea	Sclera	Cornea	Sclera
P_eff_ 10^7^ ± S.D 10^7^ Sink conditions	22.6 ± 5.52 ^a^	42.9 ± 7.38 ^a^	37.3 ± 10.5 ^b^	14.5 ± 1.48 ^b^	16.5 ± 1.81 ^b^	9.23 ± 2.01 ^b^
P_eff_ 10^7^ ± S.D 10^7^ Sink and non-sink conditions	25.6 ± 5.26	46.9 ± 6.16	38.3 ± 9.65	15.7 ± 3.41	17.5 ± 2.33	10.9 ± 3.41

## Data Availability

Data is contained within the article.

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
