# Peer review of "Topical Ocular Administration of Progesterone Decreases Photoreceptor Cell Death in Retinal Degeneration Slow (rds) Mice"

_pharmaceuticals, 2022, doi:10.3390/ph15030328_

Round 1

Reviewer 1 Report

This work studied the effect of topical ocular administration of progesterone on the photoreceptor cell death in retinal degeneration slow (rds) mice. There are some issues in this manuscript that should be addressed as follows:

  • Abstract: Page 1 Line 23: The word “show” should be replaced with “showed”.
  • Introduction:
  1. The novel points in this study should be explained in the introduction section because there are other studies that proved progesterone as a promising agent for treatment of retinal disorders.

https://pubmed.ncbi.nlm.nih.gov/30654106/

https://pubmed.ncbi.nlm.nih.gov/29551993/

https://pubmed.ncbi.nlm.nih.gov/30832304/

  1. The aim of the study mentioned in the “Introduction” section should be similar to that mentioned in the “Abstract”.
  • Materials and methods:
  1. The head line “Materials and methods” is missing. Please, revise
  2. The exact source, concentrations and the catalogue numbers of the used drugs, kits and chemicals should be mentioned.
  3. The use of commercially available progesterone is not recommended. I think it is better to use progesterone in powder form and to prepare it to be suitable for topical application on the eye.
  4. The exact number of mice used in this study should be mentioned.
  5. Page 4 Line 107: Why the number of mice in each group was not constant (8 to 10 per each group)?
  6. A reference for the used doses of PG and duration of treatment should be added.
  7. A reference for the method of preparation of the ocular specimens should be added.
  8. The method of quantification of the results of the histological and immunofluorescence examination should be mentioned.
  9. I think that the histopathological examination is not sufficient. I suggest to carry out electron microscopic study of the retinal tissues because the electron microscopic changes in these tissues precede the gross changes in the histopathological examination.
  10. Statistical analysis: The p-value below which the results are considered significant should be mentioned.
  • Results:
  1. Figures 2A and 4A: The quality of this figure should be improved.
  2. A collective diagram summarizing the main findings of this study is recommended.
  • Discussion:

The discussion should be summarized to focus on analysis of the results of the present study.

  • Conclusion:

The possible implication of the findings of the present study in the clinical settings should be mentioned.

  • General comments:
  1. The manuscript should be revised by English-naïve speaker to improve the quality of the language.
  2. The manuscript should be checked regarding the grammatical errors and plagiarism.

Author Response

Many thanks for all the comments made which we feel are very appropriate. We have answered all the points raised and where necessary, changes have been incorporated to the manuscript and have been highlighted. We are convinced the quality of the manuscript is now much higher and we would like to acknowledge the time and attention spent in reading our manuscript.

Comments and Suggestions for Authors

This work studied the effect of topical ocular administration of progesterone on the photoreceptor cell death in retinal degeneration slow (rds) mice. There are some issues in this manuscript that should be addressed as follows:

Abstract:

Page 1 Line 23: The word “show” should be replaced with “showed”.

ANSWER: Following your suggestion, the word has been replaced.

Introduction:

  1. The novel points in this study should be explained in the introduction section because there are other studies that proved progesterone as a promising agent for treatment of retinal disorders.

https://pubmed.ncbi.nlm.nih.gov/30654106/

https://pubmed.ncbi.nlm.nih.gov/29551993/

https://pubmed.ncbi.nlm.nih.gov/30832304/

ANSWER: Thank you very much for your comment. These references have been incorporated to the introduction of the manuscript (lines 96-99)

  1. The aim of the study mentioned in the “Introduction” section should be similar to that mentioned in the “Abstract”.

ANSWER: The purpose of the abstract is now similar to that of the introduction (Lines 16-20).

Materials and methods:

  1. The head line “Materials and methods” is missing. Please, revise

ANSWER: Apologies for that blatant error which has been corrected. We are grateful it has been spotted.

  1. The exact source, concentrations and the catalogue numbers of the used drugs, kits and chemicals should be mentioned.

ANSWER: The reference of the kits used in this paper are: TUNEL assay in situ cell death detection kit (Roche Diagnostics, Mannheim, Germany) (11684817910), (anti-GFAP) (1:500, Dako cytomation, Denmark), Alexa Fluor 488 (1:200, Invitrogen, Life Technologies, Madrid, Spain) (Cat # A32731) and Vectashield with DAPI (H-1200) (Vector, Burlingame, USA). This information has been incorporated in section 2.2

  1. The use of commercially available progesterone is not recommended. I think it is better to use progesterone in powder form and to prepare it to be suitable for topical application on the eye.

ANSWER: Thank you for your appreciation but we are not sure how to interpret this comment. We have not used commercially available progesterone for therapeutic use. All our studies have been performed with progesterone powder which has been reconstituted in the laboratory or in the case of the mixture of cyclodextrins and progesterone, the complex which was purchased from Sigma-Aldrich as dry powder was also reconstituted in water. This information has been clarified in lines 220 and 221.

  1. The exact number of mice used in this study should be mentioned.
  2. Page 4 Line 107: Why the number of mice in each group was not constant (8 to 10 per each group)?

ANSWER: We used 8 mice in the control group and 10 in the PG-treated group. This information has now been added to the paper, lines 123-124

The uneven number of animals in both groups allow us to comply with the R for Reduction when using animals for research. We maximize the number of animals in the experimental group while keeping the number of controls to the minimum. This strategy was overseen and approved by the Committee for animal experimentation.

  1. A reference for the used doses of PG and duration of treatment should be added.

ANSWER: Unfortunately, we cannot provide a reference for the dose of PG used because as far as we know, this is the first time PG has been used for topical ocular administration. The PG dose used is stated in the line 121 of the manuscript and the duration of the treatment is explained in the line 123.

  1. A reference for the method of preparation of the ocular specimens should be added.

ANSWER: References were added on the paper, line 148.

  1. The method of quantification of the results of the histological and immunofluorescence examination should be mentioned.

ANSWER: The method of quantification and the software used for this purpose were both described in the section Material and methods from Line 151 to Line 162.

  1. I think that the histopathological examination is not sufficient. I suggest to carry out electron microscopic study of the retinal tissues because the electron microscopic changes in these tissues precede the gross changes in the histopathological examination.

ANSWER: The main purpose of this research project could be achieved by using fluorescence microscopy. We agree with the reviewer that it could be interesting to find out if the histopathological changes seen could be, to some extent detected earlier with electron microscopy. We keep it in mind and if we have the opportunity, we shall address it in further research.

  1. Statistical analysis: The p-value below which the results are considered significant should be mentioned.

ANSWER: The p-value is implicitly mentioned when we state that “Analyses were carried out using the software IBM SPSS v27 with α= 0.05”. (lines 193-199). As you know, α is the critical value that we measure p-values against. In other words, α tells us how extreme observed results must be in order to reject the null hypothesis. In practice, to be able to find out if an observed outcome is statistically significant, we compare the values of alpha and the p-value. When the p-value is less than or equal to alpha, is when one proceeds to reject the null hypothesis.

Results:

  1. Figures 2A and 4A: The quality of this figure should be improved.

ANSWER: We suspect you feel the figures are not of sufficient quality as you get them in pdf. High quality files of all figures have been uploaded to the editorial processing system.

  1. A collective diagram summarizing the main findings of this study is recommended.

ANSWER: A graphical abstract was uploaded to the editorial processing system.

Discussion:

The discussion should be summarized to focus on analysis of the results of the present study.

ANSWER: the discussion has been rewritten and reorganized according to the criteria of all reviewers. We believe that now the discussion is more understandable to the reader.

Conclusion:

The possible implication of the findings of the present study in the clinical settings should be mentioned.

ANSWER: As suggested, justification for the possible implication of the findings of the present study in clinical settings has been added to the manuscript.

General comments:

  1. The manuscript should be revised by English-naïve speaker to improve the quality of the language.

ANSWER: The manuscript has been reviewed by a native English speaker.

  1. The manuscript should be checked regarding the grammatical errors and plagiarism.

ANSWER:

The final version of the manuscript was screened for plagiarism and this reviewed version was screened again with very satisfactory results.

Reviewer 2 Report

The present article discusses the retina pharmacological effect of a solution of progesterone in β-cyclodextrins applied topically onto mice eyes with retinitis pigmentosa.

Three different histological immunofluorescence studies were done to show the protective action of progesterone in the retina. However, the new data is quite limited and the connection to the new permeability calculated values of earlier formulations, and the benefit of new calculated permeabilities are unclear. The discussion is quite limited citing only own research findings. The authors are recommended to have more extensive discussion comparing the pharmacological effect of progesterone formulated in cyclodextrin (or other formulations) from other research sources to support own hypothesis. The manuscript requires also a reorganisation. I recommend major changes for publication.

Corrections in Material and methods section:

Three different histological immunofluorescence studies were done:

TUNEL, GFAP, and DAPI staining. While the TUNEL study is described as the way to detect the dying cells in retina, there is a lack of clear explanation in materials and methods about the information provided by the other two. The authors are recommended to explain that GFAP informs about gliosis and why the gliosis is the detrimental for the retina. What does DAPI staining inform about? It should be estate that relates to the ONL and its significance.

Progesterone is formulated with cyclodextrin, and this is the formulation used in the experiments. Therefore, progesterone alone or its abbreviation PG should not be used within the manuscript but CD-PG.

The behaviour of solution with only progesterone is very different from a formulation containing cyclodextrin and progesterone, using the right term “CD-PG” will avoid misunderstandings to the reader.

Therefore, CD-PG should be used when writing CD-PG drops, CD-PG-Blank, and CD-PG-T (treated eye), CD-PG and micelle-PG permeability coefficients, topical CD-PG solution  etc… (avoiding different other versions, or later explanations once the abbreviation has been introduced). 

The authors present a new way for calculating permeability values of the various progesterone formulations.

They are recommended to prove that the equation could be use in the same setting that in reference 25, were they used a monolayer cell line, which is quite different from corneal tissue or scleral tissue. How the authors justify that this is a more relevant equation than the previous original one?. The authors should present the results of this new calculation in results section, and not in discussion. The reported permeability values from the original references versus the new ones with the new equation should be presented in a table, and insights of the benefit of these new permeability value should be provided. What does it inform?

The authors are recommended to exclude information about the insert from the table, since you are not investigating the permeability of the insert, but the progesterone included in it, that seems to be formulated also with cyclodextrin. Therefore, the permeability for this formulation should be equal to the drop of “CD-PG”, and only the controlled release constant from the insert is playing a role. However, I am not sure how this could be demonstrated either than in in vivo setting where blinking, drainage of the excess of the drop volume, lacrimation, and conjunctival clearance are expected to have a big impact in the drug elimination, a part of the tear flow.

Corrections in Results section:

In line 194, the authors wrote: This data also demonstrate that topically applied PG protects photoreceptors and delays cell death. Pleas rewrite the sentence as “This data also demonstrate that topically applied CD-PG protects photoreceptors and delays cell death in this mice animal model”.

In line 224, the authors wrote: although the maximal amount of cell death in the rds mouse model occurs between PD17 and PD21 [26], a significant decrease in the amount of ONL cell rows would have occurred later in time and therefore has not been detected in our experimental model.

However, the authors do not know this. I would recommend reorganising the sentence such as:

A possible explanation for not observing a significant decrease in the amount of ONL cell rows could be… which was observed by the research group from reference 26.

Corrections in Discussion section:

However, the discussion section is quite weak lacking more evidence to discuss and comparing only with own previous findings (references 7 and 15). In some parts of the text, the authors present the oral progesterone results first, when it should be presented first the results obtained in this manuscript, and compare to the published data. Is there any other research group that has been investigating this drug? Has been investigated in some other animal models?

In line 318, the authors wrote: This has been demonstrated by biodistribution studies in pig eyes performed by our research group, in which, at the same initial concentration, the solution delivered 2.63 ± 1.08 μg of PG to neuroretina compared with 66.32 ± 29.52 μg delivered by 320 the insert [data pending publication]. Is this in vivo study? Is this CD_PG? Could you describe the setting?

This needs more description.

Author Response

Many thanks for all the comments made which we feel are very appropriate. We have answered all the points raised and where necessary, changes have been incorporated to the manuscript and have been highlighted. We are convinced the quality of the manuscript is now much higher and we would like to acknowledge the time and attention spent in reading our manuscript.

The present article discusses the retina pharmacological effect of a solution of progesterone in β-cyclodextrins applied topically onto mice eyes with retinitis pigmentosa.

Three different histological immunofluorescence studies were done to show the protective action of progesterone in the retina. However, the new data is quite limited and the connection to the new permeability calculated values of earlier formulations, and the benefit of new calculated permeabilities are unclear. The discussion is quite limited citing only own research findings. The authors are recommended to have more extensive discussion comparing the pharmacological effect of progesterone formulated in cyclodextrin (or other formulations) from other research sources to support own hypothesis. The manuscript requires also a reorganisation. I recommend major changes for publication.

Corrections in Material and methods section:

Three different histological immunofluorescence studies were done:

TUNEL, GFAP, and DAPI staining. While the TUNEL study is described as the way to detect the dying cells in retina, there is a lack of clear explanation in materials and methods about the information provided by the other two. The authors are recommended to explain that GFAP informs about gliosis and why the gliosis is the detrimental for the retina. What does DAPI staining inform about? It should be estate that relates to the ONL and its significance.

ANSWER: We cannot but agree that your comments are most appropriate. A short explanation about the information each of the techniques used (TUNEL, GFAP and DAPI) has been added to M&M section. Your recommendation about explaining the effect of gliosis has been followed up and a sentence has been added to the discussion (lines 307-316)

Progesterone is formulated with cyclodextrin, and this is the formulation used in the experiments. Therefore, progesterone alone or its abbreviation PG should not be used within the manuscript but CD-PG.

The behaviour of solution with only progesterone is very different from a formulation containing cyclodextrin and progesterone, using the right term “CD-PG” will avoid misunderstandings to the reader.

Therefore, CD-PG should be used when writing CD-PG drops, CD-PG-Blank, and CD-PG-T (treated eye), CD-PG and micelle-PG permeability coefficients, topical CD-PG solution etc… (avoiding different other versions, or later explanations once the abbreviation has been introduced). 

ANSWER: We fully agree that this will avoid misunderstanding and therefore the abbreviation CD-PG has been used throughout the manuscript.

The authors present a new way for calculating permeability values of the various progesterone formulations.

They are recommended to prove that the equation could be use in the same setting that in reference 25, were they used a monolayer cell line, which is quite different from corneal tissue or scleral tissue. How the authors justify that this is a more relevant equation than the previous original one? The authors should present the results of this new calculation in results section, and not in discussion. The reported permeability values from the original references versus the new ones with the new equation should be presented in a table, and insights of the benefit of these new permeability value should be provided. What does it inform?

ANSWER: Thank you very much for your comment. The results of recalculation of permeability values from the original references together with those resulting from applying the new one have been moved to the Results section. A new table has been added following your expert advice in which the PG permeability values from the cited references and those calculated with the new equation (Table 1, lines 267 and 268) which is valid under sinking or non-sinking conditions is shown. Because in the diffusion experiments performed with the inserts, sinking conditions were not achieved, we considered less accurate to calculate the permeability coefficients assuming constant concentrations in the donor compartment. However, as can be seen in table 1, the average permeability coefficient obtained with this mathematical treatment does not differ significantly from the value obtained by assuming that the concentration in the donor compartment remains constant (p> 0.05) in any of the studied settings and therefore this should serve as demonstration that the equation can be used as efficiently in monolayers cell lines or in ocular tissues.

The authors are recommended to exclude information about the insert from the table, since you are not investigating the permeability of the insert, but the progesterone included in it, that seems to be formulated also with cyclodextrin. Therefore, the permeability for this formulation should be equal to the drop of “CD-PG”, and only the controlled release constant from the insert is playing a role. However, I am not sure how this could be demonstrated either than in in vivo setting where blinking, drainage of the excess of the drop volume, lacrimation, and conjunctival clearance are expected to have a big impact in the drug elimination, a part of the tear flow.

ANSWER: As can be seen in Figure 5, coefficients of permeability are different in CD-PG solution and in CD-PG insert, for both corneal and scleral membranes, so the controlled release constant from the insert is playing a role.

We had hitherto unpublished data in which the role of tear clearance was studied in ex vivo PG biodistribution experiments from CD-PG formulated inserts (paper in process). Although we cannot describe it in this paper, we have demonstrated that no significant differences were obtained between PG concentrations in neuroretina in an ex vivo pig eye model with and without tear clearance.

Corrections in Results section:

In line 194, the authors wrote: This data also demonstrate that topically applied PG protects photoreceptors and delays cell death. Pleas rewrite the sentence as “This data also demonstrate that topically applied CD-PG protects photoreceptors and delays cell death in this mice animal model”.

ANSWER: The suggested change has been implemented.

 In line 224, the authors wrote: although the maximal amount of cell death in the rds mouse model occurs between PD17 and PD21 [26], a significant decrease in the amount of ONL cell rows would have occurred later in time and therefore has not been detected in our experimental model.

However, the authors do not know this. I would recommend reorganising the sentence such as: A possible explanation for not observing a significant decrease in the amount of ONL cell rows could be… which was observed by the research group from reference 26.

ANSWER: Your comments have been taken into account; the paragraph has been rewritten following your recommendations. We believe that it is now much more understandable for the reader.

Corrections in Discussion section:

 However, the discussion section is quite weak lacking more evidence to discuss and comparing only with own previous findings (references 7 and 15). In some parts of the text, the authors present the oral progesterone results first, when it should be presented first the results obtained in this manuscript, and compare to the published data. Is there any other research group that has been investigating this drug? Has been investigated in some other animal models?

ANSWER: Thanks a lot for your suggestion, this part of the manuscript has been rewritten at lines 322 to 329.

In line 318, the authors wrote: This has been demonstrated by biodistribution studies in pig eyes performed by our research group, in which, at the same initial concentration, the solution delivered 2.63 ± 1.08 μg of PG to neuroretina compared with 66.32 ± 29.52 μg delivered by 320 the insert [data pending publication]. Is this in vivo study? Is this CD_PG? Could you describe the setting? This needs more description.

ANSWER: this section has been rewritten considering your comments. It should now be better understood by the reader.

“This has been demonstrated by ex vivo biodistribution studies in pig eyes performed by our research group, in which, at the same initial concentration of CD-PG. The distribution through ocular tissues of different formulations of PG was studied at the same initial concentration of CD-PG. Different formulations were prepared with CD-PG in their composition drops (1mg/mL of PG), corneal/scleral insert (325.7 μg/cm2) and 3 scleral inserts (32.57, 325.7 and 3257 μg/cm2). Using whole porcine eyes, the different formulations were administered topically to the ocular surface. After freezing the eye for 12 hours at -80 °C, all tissues were dissected, and PGs were extracted from each tissue fraction in acetonitrile for 12 hours. Quantification of PGs in each tissue was performed by UHPLC-MS/MS. The solution delivered 2.63 ± 1.08 μg of PG to neuroretina compared with 66.32 ± 29.52 μg delivered by the insert [data pending publication].”

This information was included in the manuscript, lines 322 to 329.

Reviewer 3 Report

The paper is relevant and contributes to the body 
of knowledge concerning RP therapies. 
The language is very structured, concise and easy to understand.
It is a quality work presenting well both results and arguments. 

Author Response

Many thanks for your comments. We are delighted you find the paper interesting and easy to follow and also that you feel our work is sufficiently good.

Round 2

Reviewer 2 Report

The authors have well addressed my questions and recommendations.

I am a bit concerned about the non-published data that is now explained in too much detail. To avoid problems with the journal that is going to publish your new data, I suggest just the below text or something similar:

“This has been demonstrated in ex vivo biodistribution studies in pig eyes performed by our research group, with different formulations of CD-PG (drops and corneal, scleral inserts of different sizes) at the same initial concentration. They were administered topically to the ocular surface of whole pigs’ eyes, during x time experiment. PG levels were quantified in various ocular tissues. In the neuroretina, PG concentration was X for the sol CD-PG, and XX from one of the inserts [data pending publication]”.

Please also add in the main conclusions that the clinical implications may be relevant BUT further studies in larger animals are warranty.

One should be critical when considering translation from mice to humans, it is not straightforward due to immense anatomical differences which may very well result in much lower levels of drug concentration in the human retina.

The authors are recommended to have a second read of the manuscript to add some spaces and parenthesis that are missing in the text, and CD-PG Blank in conclusions...etc.

I consider the manuscript ready for publication after including these last small changes.

Author Response

Thank you very much for your comments. Following your suggestion, we have changed the text and corrected small errors throughout the manuscript. Changes in the manuscript are highlighted using tracked-changes as requested.

Regards.